# Associations of n-3, n-6 Fatty Acids Intakes and n-6:n-3 Ratio with the Risk of Depressive Symptoms: NHANES 2009–2016

**DOI:** 10.3390/nu12010240

**Published:** 2020-01-16

**Authors:** Ronghui Zhang, Jing Sun, Yan Li, Dongfeng Zhang

**Affiliations:** Department of Epidemiology and Health Statistics, The School of Public Health of Qingdao University, No. 308 Ningxia Road, Qingdao 266021, China; ZhangRH9634@163.com (R.Z.); sunjing1011@163.com (J.S.); 17861431130@163.com (Y.L.)

**Keywords:** depressive symptoms, n-3 fatty acids, n-6 fatty acids, n-6:n-3 ratio, dose-response

## Abstract

Many studies have explored the association between n-3 fatty acids and depression, but research on the associations of n-6 fatty acids and n-6:n-3 ratio with depression is more scarce, and the results are controversial. Therefore, we conducted this cross-sectional study to explore the associations of n-3 and n-6 fatty acid intakes and n-6:n-3 ratio with the risk of depressive symptoms using data from National Health and Nutrition Examination Survey (NHANES) 2009–2016. Dietary data on n-3 and n-6 fatty acids were obtained through two 24-h dietary recall interviews, and were adjusted by energy. Depressive symptoms were measured by PHQ-9 (nine-item Patient Health Questionnaire). We applied logistic regression and restricted cubic spline models to assess the relationships of n-3 and n-6 fatty acids intake and n-6:n-3 ratio with the risk of depressive symptoms. A total of 17,431 individuals over 18 years old were enrolled in this study. In the multivariate-adjusted model 2, compared with the lowest category, the highest odd ratios (ORs) with 95% confidence intervals (CIs) for n-3 fatty acid intake and n-6:n-3 ratio were 0.71 (0.55–0.92) and 1.66 (1.10–2.50), and middle OR (95% CI) for n-6 fatty acid intake was 0.72 (0.56–0.92), respectively. Our study suggests that n-3 and n-6 fatty acids intake were inversely associated with the risk of depressive symptoms, while the n-6:n-3 ratio was positively associated with the risk of depressive symptoms.

## 1. Introduction

According to the WHO (World Health Organization), more than 300 million people suffer from depression worldwide [1]. Moreover, depression has been ranked as one of the largest contributors to global disability and the risk of mortality [2]. Many studies have suggested that depression is often comorbid with many chronic diseases [3,4,5], which may gradually worsen people’s health. Thus, it is indispensable to investigate the adjustable risk factors to prevent depression.

Epidemiologic studies have shown that depression is related to genetic and environmental factors, especially dietary factors [6,7,8,9]. For instance, vegetables [10], fruits [10], fish [11], and dietary fiber [12] have been reported to reduce the risk of depression, and some nutrients [13,14] (e.g., magnesium, zinc, iron, copper, and selenium) can also reduce the risk of depression.

As essential nutrients for human body, polyunsaturated fatty acids (mainly n-3 and n-6 fatty acids) have also been reported to be associated with depression [15,16,17,18]. Observational and experimental studies have indicated that a higher consumption of n-3 fatty acids is associated with lower risk of depression [17,18,19,20,21,22,23,24,25,26,27,28], suggesting that n-3 fatty acids have a protective effect on depression. However, few studies have explored the association between n-6 fatty acids and depression, and the results are inconsistent [16,17,18,29]. The Nurses’ Health Study [16] performed among American women indicated that a higher intake of linoleic acid (the major component of n-6 fatty acids) may increase the risk of depression, while the Japan Public Center-based Prospective Study (JPHC study) [17] and Shika’s study [18] reported no association in American and Japanese populations. Moreover, studies on the association of n-6:n-3 ratio and depressive symptoms are also controversial [16,17,18,23,30,31]. A cross-sectional study [23] and two cohort studies [16,30] suggested a positive association, whereas several other studies reported no association [17,18,31]. Furthermore, few studies have investigated the dose–response relationship between n-3 fatty acids and depression [17], and no study has explored the dose–response relationship between dietary n-6 fatty acids and n-6:n-3 ratio and depression. Therefore, we explored the associations and dose–response relationships of total n-3 fatty acids, total n-6 fatty acids, and n-6:n-3 ratio with the risk of depressive symptoms in US adults based on data from the National Health and Nutrition Examination Survey (NHANES) 2009–2016.

## 2. Materials and Methods

### 2.1. Study Population

The NHANES is an ongoing, 2-year-cycle program under administration of the Centers for Disease Control and Prevention (CDC) of the US. The detailed procedures of NHANES data collection are described in the literature [32]. The study protocol is approved by National Centers for Health Statics (NCHS), and all participants gave informed consent. In this study, public data from four cycles of NHANES (2009–2010, 2011–2012, 2013–2014, and 2015–2016) were used. A total of 40,439 people participated in the NHANES from 2009 to 2016, and our analyses were restricted to 24,496 individuals over 18 years old. Among them, we excluded 325 pregnant and lactating women, 3518 participants with incomplete depression questionnaires, 3156 individuals with incomplete 24-h recall data, and 66 individuals with extreme total energy intake (<500 kcal/day for both men and women, >5000 kcal/day for women, and >8000 kcal/day for men). Finally, a total of 17,431 participants were included in the present study (Figure 1).

### 2.2. Assessment of Depressive Symptoms

Depressive symptoms were assessed by the PHQ-9 (nine-item Patient Health Questionnaire) which integrates DSM-IV depression diagnostic criteria and was found to be a reliable and effective screening instrument in both clinical and research settings [33,34]. The total scores of PHQ-9 range from 0 to 27, and a score of 10 was used as the cut-off point to identify depression according to Kroenke et al. [35].

### 2.3. Dietary n-3 and n-6 Fatty Acid Intakes

Dietary n-3 and n-6 fatty acids intake were assessed by two 24-h dietary recall interviews. The subtypes of n-3 and n-6 fatty acids in this study are consistent with our previous studies [36]. The daily average n-3 and n-6 fatty acids intake were adjusted by energy.

### 2.4. Covariates

The following covariates were included in this study: age, gender, race, marital status, education level, annual household income, smoking status, alcohol consumption status, work activity, recreational activity, body mass index (BMI), hypertension, diabetes, coronary heart disease, and energy intake. The classifications of covariates were based on our previous studies [13,14,37] and are shown in Appendix A.

### 2.5. Statistical Analysis

Kolmogorov–Smirnov normality tests were used to test the normality of continuous variables. The normally distributed variables were described by mean ± standard deviation, and non-normally distributed variables were described by median (interquartile range). According to the characteristics of variables, *t*-tests or chi-square tests were applied to compare the differences between the depressive symptoms group and the non-depressive-symptoms group. The adjusted dietary n-3 and n-6 fatty acids intake were divided into three groups based on tertiles; the lowest tertile was the reference. In view of the large number of people whose dietary n-6:n-3 ratio was less than 10 in our study, and because studies have proved that the average value of dietary n-6:n-3 ratio is about 15 [38], we divided the n-6:n-3 ratio into three groups (group 1: <10; group 2: ≥10 to 15; group 3: >15), where group 1 was the reference.

Binary logistic regression analyses were conducted to explore the associations between dietary n-3 and n-6 fatty acids, n-6:n-3 ratio, and the risk of depressive symptoms. Only age and gender were adjusted in model 1, and all the covariates were adjusted in model 2. In addition, stratified analyses were performed to test whether these associations differed by age and gender. To further explore the dose–response relationships between dietary n-3 and n-6 fatty acids intake and n-6:n-3 ratio and the risk of depressive symptoms, we applied restricted cubic spline with three knots at the 5^th^, 50^th^, and 95^th^ percentiles of the exposure distribution in the multivariate-adjusted model 2. In order to perform nationally representative estimates, the analyses were weighted in this study. All statistical analyses were conducted using Stata 15.0. A two-sided *p*-value less than 0.05 was considered statistically significant.

## 3. Results

The characteristics of NHANES participants by depressive symptoms are shown in Table 1. The prevalence of depressive symptoms was 8.87%. Compared with the non-depressive-symptoms group, the participants with depressive symptoms tended to be older, women, single or living alone, obese, smokers, with lower education level, lower income, and lower recreation activity. The prevalence of depressive symptoms was high in the subjects with hypertension and diabetes. Moreover, the energy and n-3 and n-6 fatty acids intake in participants with depressive symptoms were lower than those without depressive symptoms, while the n-6:n-3 ratio in in participants with depressive symptoms was higher than those without depressive symptoms.

The odds ratios (ORs) with 95% confidence intervals (CIs) of depressive symptoms with n-3 and n-6 fatty acids intake and n-6:n-3 ratio are shown in Table 2. In the unadjusted model of binary logistic regression analyses, the ORs (95% CIs) were 0.67 (0.55–0.82), 0.78 (0.65–0.95), and 1.43 (1.08–1.91) for the highest group versus the reference, respectively. In model 1, n-3 and n-6 fatty acids intake and n-6:n-3 ratio were still associated with the risk of depressive symptoms. In model 2, compared with the reference, the ORs of highest group of n-3 fatty acids and n-6:n-3 ratio were 0.70 (0.55–0.92) and 1.66 (1.10–2.50), and the ORs of second group of n-6 fatty acids was 0.72 (0.56–0.92).

In stratified analyses by age and gender, the results are shown in Table 3 and Table 4, respectively. In stratified analyses by age, n-3 and n-6 fatty acids intake and n-6:n-3 ratio were associated with the risk of depressive symptoms for participants over 60 years old. The ORs (95%CIs) in model 2 were 0.42 (0.28–0.61), 0.42 (0.29–0.62), and 3.83 (1.53–9.54), respectively. In stratified analyses by gender, n-3 fatty acids intake was inversely associated with the risk of depressive symptoms among men, and the n-6:n-3 ratio was positively associated with the risk of depressive symptoms among women; the corresponding ORs (95%CIs) were 0.57 (0.36–0.91) and 1.84(1.05–3.22) in model 2.

The results of dose–response relationships are shown in Figure 2, Figure 3 and Figure 4. There was a U-shaped association between n-3 fatty acids intake and the risk of depressive symptoms (*p*
_for nonlinearity_ = 0.006). When the n-3 fatty acids intake was 1.1 mg/kcal/day, the OR value tended to the lowest (OR: 0.45; 95% CI: 0.29–0.70), and there was no significant association between the risk of depressive symptoms and n-3 fatty acids intake when the consumption was beyond 1.8 mg/kcal/day (OR: 0.62; 95% CI: 0.38–1.01). Similarly, a U-shaped association between n-6 fatty acids intake and the risk of depressive symptoms was also found. When the n-6 fatty acids intake was 8.8 mg/kcal/day, the OR value tended to the lowest (OR: 0.59; 95% CI: 0.37–0.94), and there was no significant association between the risk of depressive symptoms and n-6 fatty acids intake when the consumption was beyond 11.9 mg/kcal/day (OR: 0.66; 95% CI: 0.43–1.01). However, the n-6:n-3 ratio was linearly positively associated with the risk of depressive symptoms (*p*
_for nonlinearity_ = 0.069).

## 4. Discussion

In this study, we found that the intakes of n-3 and n-6 fatty acids were inversely associated with the risk of depressive symptoms, and dietary n-6:n-3 ratio was positively associated with the risk of depressive symptoms, and these associations were stronger among the elderly. Furthermore, the dose–response relationship analysis indicated that the n-3 and n-6 fatty acids intake had nonlinear and U-shaped associations with the risk of depressive symptoms, while n-6:n-3 ratio was linearly positively associated with the risk of depressive symptoms.

Our findings about n-3 fatty acids intake are consistent with other studies, including cross-sectional [18,23,24], prospective cohort [17,25,26,27], meta-analysis [19,28] and experimental studies [20]. The results from a cross-sectional study [23] indicated that higher n-3 fatty acids intake was inversely associated with the risk of evaluated depressive symptoms. Chika Horikawa et al. [27] also found the protective effect of n-3 fatty acids on depression through an established cohort. Moreover, a recent meta-analysis [19] explored the dose–response relationship between n-3 fatty acids and depression, and a reverse J-shaped effect was found. The inverse association between n-3 fatty acids and the risk of depression could be explained by several possibilities. First, research has shown that depression is related to inflammatory response [39], and n-3 fatty acids possess anti-inflammatory properties that reduce the risk of depression [40]. Second, n-3 fatty acids promote neurotransmitter binding and intracellular signaling [41].

We also found that n-6 fatty acids intake was inversely associated with the risk of depressive symptoms, which was inconsistent with previous studies. The NHANES Epidemiologic Followup Study (NHEFS) [29] found the opposite results: greater n-6 fatty acids intake was positively associated with severe depressive symptoms among men, while Shika’s study [18] in an elderly Japanese population indicated that the intake of n-6 fatty acids was not associated with depression. However, the reasons for the different outcomes remain to be explored, and the mechanism of this association is unclear. Therefore, further research is required to explore the relationship between n-6 fatty acids intake and the risk of depression.

In addition, a positive association between n-6:n-3 ratio and depressive symptoms was found in our study, which is consistent with several cohort studies [16,23,30]. However, other studies reported no association between n-6:n-3 ratio and depression [17,18]. Although the mechanisms between n-6:n-3 ratio and the increased risk of depression are not fully understood, several possibilities have been proposed. Primarily, a high n-6:n-3 ratio promotes the production of arachidonic acid (AA) derived from n-6 eicosanoids, and AA has been shown to increase the production of proinflammatory factors, which may increase the incidence of depression [42]. Another possible mechanism is that a high n-6:n-3 ratio is related to catecholaminergic or serotonergic neurotransmission [43].

There are several notable strengths of our study. Firstly, we simultaneously explored the associations of n-3 and n-6 fatty acids intake and n-6:n-3 ratio with the risk of depressive symptoms, and the dose–response relationships were also investigated. Secondly, we explored age and gender differences in the association between n-3 and n-6 fatty acids intake, n-6:n-3 ratio, and the risk of depressive symptoms, respectively. Finally, we used a large nationally representative sample that could increase the statistical power and provide a more reliable and accurate result.

However, there are several limitations to our study. First, since our study was cross-sectional, it was difficult to determine causality. Second, depressive symptoms were assessed by the PHQ-9 self-reported scale, and misclassification bias could not be completely avoided, which might affect the results. Third, dietary data were obtained through two 24-h dietary recall interviews, which may have uncertainties in estimating the long-term average consumption. However, some studies have shown that two 24-h recalls are sufficient to assess the daily dietary intake [44].

## 5. Conclusions

Our study indicates that the intakes of n-3 and n-6 fatty acids were inversely associated with the risk of depressive symptoms, and the n-6:n-3 ratio was positively associated with the risk of depressive symptoms. Further research is needed to verify whether these relationships are consistent.

## Figures and Tables

**Figure 1 nutrients-12-00240-f001:**
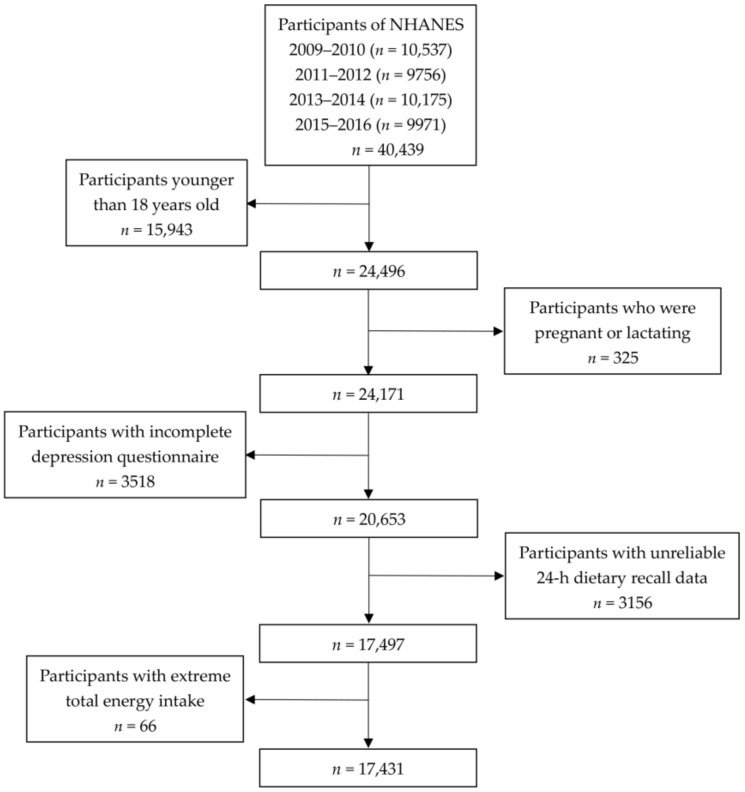
Flow chart of the screening process for the selection of eligible participants.

**Figure 2 nutrients-12-00240-f002:**
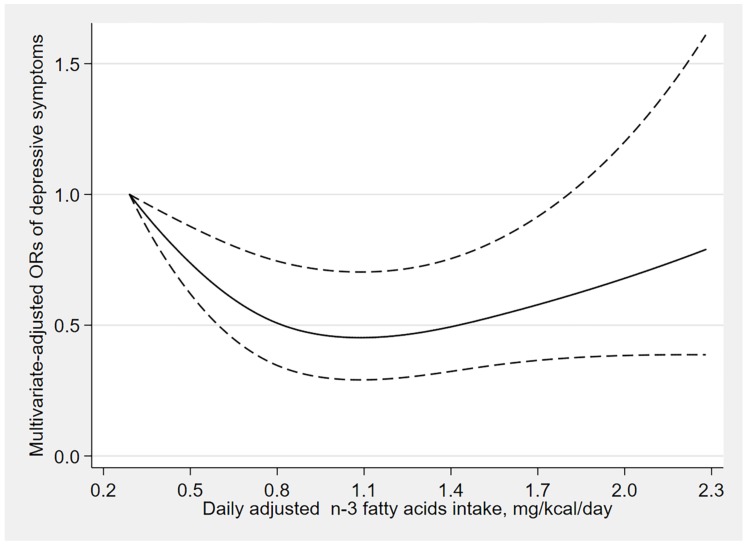
Dose–response relationship between n-3 fatty acids intake and the risk of depressive symptoms. The solid line represents the OR values and dashed lines represent the 95% confidence intervals.

**Figure 3 nutrients-12-00240-f003:**
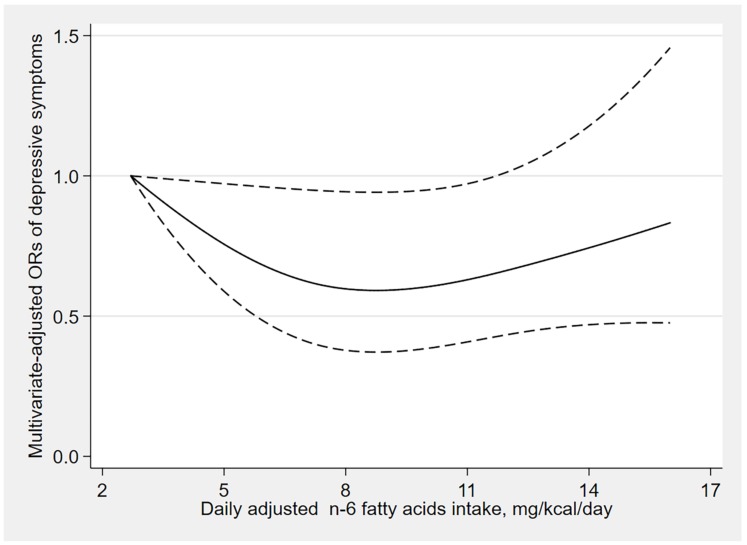
Dose–response relationship between n-6 fatty acids intake and the risk of depressive symptoms. The solid line represents the OR values and dashed lines represent the 95% confidence intervals.

**Figure 4 nutrients-12-00240-f004:**
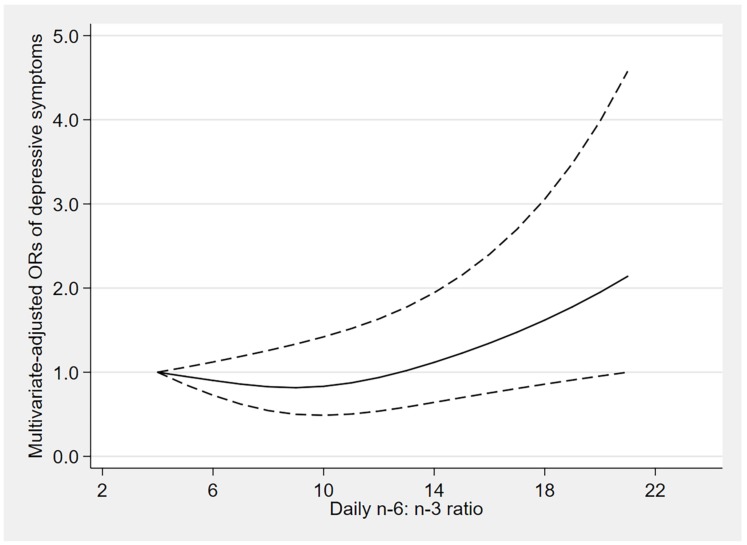
Dose–response relationship between n-6:n-3 ratio and the risk of depressive symptoms. The solid line represents the OR values and dashed lines represent the 95% confidence intervals.

**Table 1 nutrients-12-00240-t001:** Characteristics of National Health and Nutrition Examination Survey (NHANES) participants by depressive symptoms.

	With Depressive Symptoms (PHQ-9 ≥ 10)	Without Depressive Symptoms (PHQ-9 < 10)	*p*-Value
**Number of subjects (%)**	1546 (8.87)	15,885 (91.13)	
**Age group (%) ^1^**			<0.01
18–44 years	622 (40.2)	6964 (43.8)	
44–59 years	480 (31.0)	3724 (23.4)	
≥60 years	444 (28.7)	3613 (22.7)	
**Sex (%) ^1^**			<0.01
Male	553 (35.8)	8002 (50.4)	
Female	993 (64.2)	7883 (48.6)	
**Race (%) ^1^**			<0.01
Mexican American	232 (15.0)	2329 (14.6)	
Other Hispanic	214 (13.8)	1574 (9.9)	
Non-Hispanic White	648 (41.9)	6789 (42.7)	
Non-Hispanic Black	334 (21.6)	3392 (21.4)	
Other race	118 (7.6)	1810 (11.4)	
**Marital status (%) ^1^**			<0.01
Married/Living with partner	650 (42.0)	9135 (57.5)	
Windowed/Divorced/Separated/Never married	838 (54.2)	5875 (37.0)	
**Education level (%) ^1^**			<0.01
Below high school	527 (34.1)	3379 (21.3)	
High school	370 (23.9)	3669 (23.1)	
Above high school	649 (42.0)	8825 (55.6)	
**Household income (%) ^1^**			<0.01
<$20,000	579 (37.5)	2948 (18.6)	
$20,000–$44,999	529 (34.2)	5003 (31.50)	
$50,000–$74,999	208 (13.5)	2946 (18.5)	
≥$75,000	162 (10.5)	4280 (26.9)	
**Body mass index (%) ^1^**			<0.01
<25 kg/m^2^	377 (24.4)	4740 (29.8)	
25 to <30 kg/m^2^	365 (23.6)	5201 (32.7)	
≥30 kg/m^2^	783 (50.6)	5801 (36.5)	
**Smoking at least 100 cigarettes in life (%) ^1^**			<0.01
Yes	891 (57.6)	6473 (40.7)	
No	627 (40.6)	8961 (56.4)	
**Have at least 12 alcoholic drinks/year (%) ^1^**			0.175
Yes	1117 (72.3)	11,196 (70.5)	
No	410 (26.5)	4460 (28.1)	
**Work activity (%) ^1^**			0.124
Vigorous activity	294 (19.0)	2982 (18.8)	
Moderate activity	287 (18.6)	3420 (21.5)	
Other	963 (62.3)	9477 (59.7)	
**Recreation activity (%) ^1^**			<0.01
Vigorous activity	178 (11.5)	3852 (24.2)	
Moderate activity	3231 (20.9)	4360 (27.4)	
Other	1045 (67.6)	7669 (48.3)	
**Hypertension (%) ^1^**			<0.01
Yes	732 (47.3)	5519 (34.7)	
No	812 (52.5)	10,352 (65.2)	
**Diabetes (%) ^1^**			<0.01
Yes	316 (20.4)	1911 (12.0)	
No	1226 (79.3)	13,968 (87.9)	
**Coronary heart disease (%) ^1^**			<0.01
Yes	961 (62.2)	10,812 (68.1)	
No	171 (11.1)	924 (5.8)	
**Total energy intake (kcal/d) ^2^**	1952.5 (814.4)	2049.8 (799.4)	<0.01
**Adjusted n-3 fatty acid intake (mg/kcal/day) ^2^**	0.83 (0.38)	0.89 (0.41)	<0.01
**Adjusted n-6 fatty acid intake (mg/kcal/day) ^2^**	7.46 (2.83)	7.77 (2.73)	<0.01
**n-6:n-3 ratio ^2^**	9.72 (3.51)	9.43 (3.37)	<0.05

^1^ Chi-square and ^2^
*t*-tests were applied to compare the differences between the depressive-symptoms group and the non-depressive-symptoms group. PHQ-9: nine-item Patient Health Questionnaire.

**Table 2 nutrients-12-00240-t002:** The ORs (95% CIs) for depressive symptoms by adjusted dietary n-3 and n-6 fatty acids intake and n-6:n-3 ratio, NHANES 2009–2016 (*n* = 17,431).

	Crude	Model 1	Model 2
**Adjusted n-3 fatty acids intake (mg/kcal/day)**			
<0.68	1.00	1.00	1.00
0.68 to <0.95	0.70 (0.58–0.85) **	0.68 (0.56–0.82) **	0.70 (0.55–0.89) **
≥0.95	0.67 (0.55–0.82) **	0.64 (0.52–0.78) **	0.71 (0.55–0.92) *
**Adjusted n-6 fatty acids intake (mg/kcal/day)**			
<6.39	1.00	1.00	1.00
6.39 to <8.57	0.68 (0.56–0.82) **	0.66 (0.55–0.80) **	0.72 (0.56–0.92) *
≥8.57	0.78 (0.65–0.95) *	0.75 (0.62–0.90) **	0.84 (0.66–1.07)
**n-6:n-3 ratio**			
< 10	1.00	1.00	1.00
10 to <15	1.19 (0.96–1.49)	1.19 (0.96–1.49)	1.15(0.86–1.55)
≥15	1.43 (1.08–1.91) *	1.47 (1.10–1.96) **	1.66(1.10–2.50) *

* *p <* 0.05; ** *p <* 0.01.

**Table 3 nutrients-12-00240-t003:** The ORs (95%CIs) of depressive symptoms by adjusted dietary n-3 and n-6 fatty acid intakes and n-6:n-3 ratio, stratified by age, NHANES 2009–2016 (*n* = 17,431).

	18 *≤* Age < 45 Years			45 *≤* Age < 60 Years			Age *≥* 60 Years		
	Crude	Model 1	Model 2	Crude	Model 1	Model 2	Crude	Model 1	Model 2
**Adjusted n-3 fatty acids intake (mg/kcal/day)**									
<0.68	1.00	1.00	1.00	1.00	1.00	1.00	1.00	1.00	1.00
0.68 to <0.95	0.61 (0.48–0.77) **	0.58 (0.46–0.74) **	0.74 (0.54–1.02)	0.75 (0.51–1.12)	0.73 (0.50–1.08)	0.69 (0.43–1.10)	0.85 (0.53–1.35)	0.83 (0.52–1.33)	0.58 (0.35–0.95) *
≥0.95	0.66 (0.49–0.90) **	0.62 (0.46–0.84) *	0.82 (0.57–1.18)	0.82 (0.59–1.14)	0.77 (0.56–1.07)	0.78 (0.50–1.19)	0.53 (0.37–0.76)	0.50 (0.35–0.72) **	0.42 (0.28–0.61) **
**Adjusted n-6 fatty acids intake (mg/kcal/day)**									
<6.39	1.00	1.00	1.00	1.00	1.00	1.00	1.00	1.00	1.00
6.39 to <8.57	0.81 (0.60–1.09)	0.78 (0.58–1.05)	0.95 (0.68–1.33)	0.70 (0.50–0.98) *	0.69 (0.49–0.97) *	0.67 (0.44–1.01)	0.42 (0.29–0.62) **	0.42 (0.28–0.61) **	0.42 (0.27–0.64) **
≥8.57	0.79 (0.59–1.06)	0.75 (0.56–1.01)	0.94 (0.65–1.35)	0.89 (0.64–1.25)	0.85 (0.61–1.20)	0.99 (0.66–1.50)	0.61 (0.42–0.88) *	0.58 (0.40–0.84) **	0.42 (0.29–0.62) **
**n-6:n-3 ratio**									
<10	1.00	1.00	1.00	1.00	1.00	1.00	1.00	1.00	1.00
10 to <15	1.24 (0.96–1.60)	1.25 (0.96–1.62)	1.26 (0.90–1.77)	1.01 (0.68–1.51)	1.01 (0.68–1.50)	0.96 (0.57–1.64)	1.36 (0.84–2.20)	1.42 (0.88–2.30)	1.13 (0.65–1.96)
≥15	1.45 (0.91–2.31)	1.49 (0.93–2.39)	1.33 (0.73–2.42)	1.22 (0.67–2.22)	1.25 (0.70–2.26)	1.51 (0.70–1.78)	1.74 (0.79–3.86)	1.83 (0.81–4.12)	3.83 (1.53–9.54) **

* *p <* 0.05; ** *p <* 0.01.

**Table 4 nutrients-12-00240-t004:** The ORs (95%CIs) of depressive symptoms by adjusted dietary n-3 and n-6 fatty acids intake and n-6:n-3 ratio, stratified by gender. NHANES 2009–2016 (*n* = 17,431).

	Men			Women		
	Crude	Model 1	Model 2	Crude	Model 1	Model 2
**Adjusted n-3 fatty acids intake (mg/kcal/day)**						
<0.68	1.00	1.00	1.00	1.00	1.00	1.00
0.68 to <0.95	0.56 (0.38–0.83) **	0.56 (0.38–0.83) **	0.57 (0.36–0.91) *	0.75 (0.58–0.95) *	0.76 (0.58–0.99) *	0.80 (0.56–1.14)
≥0.95	0.66 (0.48–0.92) *	0.67 (0.48–0.93) *	0.77 (0.51–1.18)	0.62 (0.49–0.79) **	0.63 (0.49–0.80) **	0.71 (0.50–1.01)
**Adjusted n-6 fatty acids intake (mg/kcal/day)**						
<6.39	1.00	1.00	1.00	1.00	1.00	1.00
6.39 to <8.57	0.57 (0.41–0.80) **	0.57 (0.41–0.80) **	0.66 (0.41–1.04)	0.73 (0.59–0.90) **	0.73 (0.59–0.90) **	0.76 (0.56–1.04)
≥8.57	0.67 (0.51–0.88) **	0.67 (0.51–0.88) **	0.82 (0.55–1.22)	0.80 (0.61–1.05)	0.81 (0.61–1.06)	0.89 (0.61–1.30)
**n-6:n-3 ratio**						
<10	1.00	1.00	1.00	1.00	1.00	1.00
10 to <15	1.29 (0.92–1.81)	1.29 (0.92–1.80)	1.39 (0.92–2.12)	1.17 (0.90–1.52)	1.14 (0.87–1.48)	1.02 (0.74–1.39)
≥15	1.56 (0.94–2.16)	1.56 (0.92–2.63)	1.43 (0.73–2.79)	1.43 (0.98–2.10)	1.41 (0.96–2.06)	1.84 (1.05–3.22) *

* *p <* 0.05; ** *p <* 0.01.

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
