# Peer review of "Associations of n-3, n-6 Fatty Acids Intakes and n-6:n-3 Ratio with the Risk of Depressive Symptoms: NHANES 2009–2016"

_nutrients, 2020, doi:10.3390/nu12010240_

Round 1
Reviewer 1 Report
This is a very interesting cross selectional study explorating the hypothetical role of n-6 polyunsaturated fatty acids on depression symptoms.
Introduction
At line 51 could you please insert the ref after: “Furthermore, few studies have investigated the dose-response relationship 51 between n-3 fatty acids and depression”
Methods
The sample is very large so it garanted a powerful statistical analysis. Methods are well done, also statistical analysis.
Also results and discussion were well done, the authors described both strengths and limits of their study, I agree with their point of view. Also tables and figures are clear. Finally the conclusion is very interesting and stimulating new studies about this issue.
Author Response
Manuscript ID: nutrients-675643
Manuscript Title: Associations of n-3, n-6 fatty acids intakes and n-6: n-3 ratio with the risk of depressive symptoms: NHANES 2009–2016
Thanks a lot. According to your suggestion, we have inserted the corresponding reference (Line 51).
Reviewer 2 Report
In this study authors suggested that n-3 and n-6 fatty acids intakes were inversely associated with the 264 risks of depressive symptoms, n-6: n-3 ratio was positively associated with the risk of depressive symptoms, 265 and these associations were stronger among the elderly. The manuscript is easy to read and understand, and the results were clearly presented.
The authors carefully check the manuscript and correct all typographical errors present in the manuscript.
Author Response
Manuscript ID: nutrients-675643
Manuscript Title: Associations of n-3, n-6 fatty acids intakes and n-6: n-3 ratio with the risk of depressive symptoms: NHANES 2009–2016
Thanks for your advice. We have checked the manuscript carefully and corrected the typographical errors thoroughly.
Reviewer 3 Report
This is an interesting and well written study. The major shortcomings are its cross-sectional design and only a 24 hour recall interview data on the participant’s diet from which the n-3 and n-6 fatty acid intake is derived. The limitations are properly discussed by the authors but these restrictions also somewhat question the reliability and novelty of the findings and hence, the conclusion.
Author Response
Manuscript ID: nutrients-675643
Manuscript Title: Associations of n-3, n-6 fatty acids intakes and n-6: n-3 ratio with the risk of depressive symptoms: NHANES 2009–2016
We thank the reviewer for the thoughtful comments, and agree with the reviewer that the 24-h recall interview has some limitations. However, in our study, the dietary intakes were assessed by two 24-h recall interviews, which, according to Knuppel et al (the reference 45), are sufficient to assess the daily dietary intake. Although the cross-sectional study was difficult to determine the causality, it can provide clues for further studies. Moreover, we used a large nationally representative sample that could increase the statistical power and provide a more reliable and accurate result.